# Solid-State Surface Patterning on Polymer Using the Microcellular Foaming Process

**DOI:** 10.3390/polym15051153

**Published:** 2023-02-24

**Authors:** Jaehoo Kim, Shin Won Kim, Byung Chul Kweon, Kwan Hoon Kim, Sung Woon Cha

**Affiliations:** Department of Mechanical Engineering, Yonsei University, 50 Yonsei-ro, Seodaemun-gu, Seoul 03722, Republic of Korea

**Keywords:** solid-state batch-foaming process, surface-patterning process, polymer–gas mixture, surface roughness, compression molding, volume expansion

## Abstract

This study proposes a novel process that integrates the molding and patterning of solid-state polymers with the force generated from the volume expansion of the microcellular-foaming process (MCP) and the softening of solid-state polymers due to gas adsorption. The batch-foaming process, which is one of the MCPs, is a useful process that can cause thermal, acoustic, and electrical characteristic changes in polymer materials. However, its development is limited due to low productivity. A pattern was imprinted on the surface using a polymer gas mixture with a 3D-printed polymer mold. The process was controlled with changing weight gain by controlling saturation time. A scanning electron microscope (SEM) and confocal laser scanning microscopy were used to obtain the results. The maximum depth could be formed in the same manner as the mold geometry (sample depth: 208.7 μm; mold depth: 200 μm). Furthermore, the same pattern could be imprinted as a layer thickness of 3D printing (sample pattern gap and mold layer gap: 0.4 mm), and surface roughness was increased according to increase in the foaming ratio. This process can be used as a novel method to expand the limited applications of the batch-foaming process considering that MCPs can impart various high-value-added characteristics to polymers.

## 1. Introduction

The microcellular-foaming process (MCP) is a representative engineering technology that is commercialized owing to the attention it receives from the industry, solving the problem of plastic consumption with three constraints: reducing plastic usage, maintaining mechanical properties, and not changing the geometrical shape of a product [1]. Previously, MCPs were primarily used in packaging, automatic interior and exterior material manufacturing, construction, and insulation [2]. However, MCPs have been widely used in high-value-added industries, such as drug delivery [3], bioscaffolds [4,5], EMI shielding [6,7], and filtration [8,9].

An MCP includes three stages. The first step is to combine a polymer and a physical blowing agent (PBA), which are separated into individual phases from a single phase under high pressure [10]. Because of gas adsorption, some property changes occur in polymers, such as tensile strength, elastic modulus, and impact strength [11]. After the first step, diffusion occurs due to Gibbs free energy in a single phase, and cell nucleation and cell growth occur owing to thermodynamic instability, which is caused by a rapid change in solubility due to a change in temperature or pressure. Additionally, the third step is the cell stabilization step, which is described in Figure 1 as the step of generating and stabilizing cells [12]. Furthermore, MCPs include a solid-state batch process, injection molding, and extraction foaming. Additionally, based on the continuity of the process, they can be divided into a solid-state batch-foaming process and the remaining processes [13]. The solid-state batch-foaming process is easy to control because all the stages are separated independently. However, the processing time to produce a single phase in the first step is very large. Thus, it is primarily used for fundamental studies or lab-scale experiments [14]. In the case of the injection-molding and extraction-molding processes, although variables should be controlled, it is a continuous process with high productivity. Therefore, it is selected and used in various industrial fields based on target requirements [15]. MCPs also can be used for overcoming shrinkage or warpage, which frequently occurs in injection-molded plastics [16]. Recently, there is research about combining MCPs with other injection-molding technology, such as in-mold decoration [17]. Furthermore, there are other controllable process parameters, such as mold temperature [18], and counterpressure [19]. By optimizing these variables, MCP injection molding has kept developing [20,21]. Although the solid-state batch process was developed 40 years ago, due to its low productivity, industrial applications are limited to McPET and McPC, which utilize changes in reactivity after foaming [22]. Injection molding and extrusion are utilized in industry and actively used in automobile interior materials and structures, whereas batch solid-state processing with insufficient application is considered a minor part of MCPs and a basic field that cannot be applied to industry. However, new studies on application processes using batch solid-states have recently emerged. A novel continuous batch-foaming process was proposed by combining an additive management process and solid-state batch foaming, resulting in effective cell delay [22]. Additionally, studies have been conducted to form a solid polymer–gas mixture through a batch process and then perform foam patterning on its surface through thermal activation [23,24]. A solid-state process performed at room temperature and pressure has the advantage of significantly reducing energy consumption [25]. In addition, because of easy-to-control foaming conditions in the batch-foaming process, it is easy to achieve target properties, such as optical reflectivity [26] and electronic characteristics [27].

This study seeks to solve the reduction in energy used in the patterning process by compression molding while maintaining resolution through gas saturation and the foaming process. First, the gas adsorption is predicted through two experimental values (weight gain) using a suggested model [28], and the experimental parameters are determined through this curve. A specimen is placed in a patterning jig manufactured using a 3D printer to perform a compression-molding process, and the force generated by volume expansion is converted into a force taking a pattern in reverse. Through this study, the process of patterning a solid-phase polymer surface is successful with multiple scales of pattern, and the procedure is named solid-state surface patterning on a polymer using a microcellular-foaming process.

## 2. Materials and Methods

### 2.1. Materials

#### 2.1.1. Specimen

In this study, polymethyl methacrylate (PMMA) (LX MMA, Yeosu City, Jeollanam-do, Republic of Korea) was used, which is known as acrylic and is a representative amorphous thermoplastic. Additionally, it is currently used as a substitute due to high resistance to wear. This study proposed a novel solid-state surface-pattern-imaging process based on volume expansion that occurred during the foaming process; a specimen with large volume expansion was more advantageous. Therefore, to increase the volume expansion, PMMA, which has a carbonyl group friendly to CO_2_ as a functional group [29], was selected as the target of the experiment, with a much higher amount of CO_2_ gas adsorption than other amorphous thermoplastics. The geometry and properties of the specimen are listed in Table 1.

#### 2.1.2. Blowing Agents

Nitrogen and CO_2_ were the primary gases used as physical blowing agents (PBAs), and the two gases were selected based on the target properties of the experiment and the cellular plastic. CO_2_ shows relatively high solubility compared to nitrogen, with low diffusivity [30]. Therefore, in the case of an injection-molding process that implements a low expansion ratio (high-density foam), nitrogen is used rather than carbon dioxide as a physical foaming agent [31]. In contrast, as described in the previous chapter, CO_2_ is used as a physical foaming agent due to conditions where gas addition should be high (purity: 99.9%; 40 L; Samhung GasTech, Seoul, Republic of Korea).

### 2.2. Experiment Setup

#### 2.2.1. Solid-State Batch-Foaming Process

A high-pressure vessel was used to form a solid-state polymer–gas mixture, and an image of the actual equipment used in this experiment is shown in Figure 2. A batch-foaming process is classified as a one-step technique, which induces thermodynamic instability through a pressure drop, or as a two-step technique, which induces thermodynamic instability through temperature [32]. In this study, cell nucleation and growth was omitted despite using the two-stage technique of quenching the polymer–gas mixture in glycerin above the glass transition temperature. Therefore, a compression force was applied to the polymer–gas mixture to induce cell nucleation and growth after producing the polymer–gas mixture through the gas saturation process. The inner and outer radii of the pressure vessel were 75 and 100 mm, the inner and outer heights were 150 and 185 mm, and the total thickness was 25 mm, respectively. The gas adsorption, or weight gain, was measured using the gravimetric method with an acuity of 0.01 g, (OHAUs, Model no. AR2130), and the expression for the measurement method can be observed in Equation (1).
(1)$$\mathrm{Weight}\;\mathrm{gain}\;\left(\%\right)=\frac{{\mathrm{Weight}}_{\mathrm{adsorption}}\;\left(g\right)-{\mathrm{Weight}}_{\mathrm{neat}}\;\left(g\right)}{{\mathrm{Weight}}_{\mathrm{neat}}\;\left(g\right)}$$

#### 2.2.2. Compression Molding

The equipment of the compression-molding machine used in this process is shown in Figure 3. The compressor connected to the compression-molding machine had a maximum air pressure of 848 kPa, 3.375 kW, and a 40 L air tank. A mold-releasing agent was not used in this research. The temperature of the top and bottom could be controlled up to 350 °C with a thermocompression molding machine. The research only conducted experiments with simple compression and did not apply heat since the research was conducted at room temperature without thermal activation. The compression-molding jig is illustrated in Figure 3b, and the inner part was 60 mm in width, 60 mm in length, and 5 mm in thickness. Even if the thickness of the specimen used in this experiment was 1.25–1.28 mm and swelling by CO_2_ was considered in a situation where foaming did not occur and a single phase of polymer–CO_2_ was formed, compression did not proceed because a gap of 3–3.5 mm occurred immediately. Therefore, for an accurate compression-molding process and surface patterning, a jig capable of fixing a specimen to the compression mold was produced using a 3D printer.

#### 2.2.3. Specimen Jig Using 3D Printer

A 3D printer was used to manufacture a jig for surface patterning in a specimen injected with PBA inside the polymer. The experimental equipment used was a 3Dwox1 instrument (Sindoh, Seoul, Republic of Korea), and 1.75 mm of PLA filament was used. Furthermore, PLA has a glass transition temperature in the range of 50–80 °C, which is unsuitable for compression molding, which gives general temperature and pressure together. However, since thermal activation was not required in this experiment, a PLA with relatively high numerical stability was used. For accurate compression, a jig that could fix the specimen was produced with a 3D printer, as illustrated in Figure 4. The 3D printer jig had an outer length of 29 mm, an outer height of 5 mm, a jig wall of 1 mm, and an inner depth of 1.3 mm. The pattern was created as a two-line model, the width of the pattern was 0.1 mm, and the depth of the pattern was 0.1 mm.

#### 2.2.4. Overall Process

In Figure 5, the entire process used in this experiment is integrated into a schematic. First, to form the polymer–gas mixture state of the specimen, the specimen was saturated using a solid-state batch-foaming process. After saturation, pressure was discharged from the pressure vessel at a constant depressurization ratio and fixed to the jig drawn with the 3D printer, and the specimen was positioned on the compression-molding jig. In the steady state where the process was sufficiently completed, each experimental variable and the depth of the pattern was checked based on several factors, including weight gain and cell morphology.

## 3. Results

### 3.1. Gas Adsorption

Gas saturation was conducted to form a polymer–gas mixture (i.e., a single phase). The experimental conditions are listed in Table 2. The microcellular-foaming process had six experimental parameters: saturation pressure, saturation temperature, saturation time, depressurization ratio, foaming temperature, and foaming time. In this study, because thermodynamic instability through thermal stimulation was not desired, the saturation temperature was set to room temperature, where the patterning process proceeded and foaming temperature and foaming time were not considered. Because the intention was to confirm the tendency of surface patterning as a function of gas concentration in the polymer, the experiment was conducted while changing only the saturation time in a fixed state. The trend of the adsorption curve was confirmed using Equation (2), which was proposed for gas adsorption in a solid polymer based on Fick’s diffusion [28].
(2)$$\frac{M_{t}-M_{0}}{M_{\infty }-M_{0}}=1-\frac{8}{\pi^{2}}\sum_{n=0}^{\infty }\frac{1}{{\left(2n+1\right)}^{2}}\exp\left(-\frac{{\left(2n+1\right)}^{2}\pi^{2}\mathrm{Dt}}{4L^{2}}\right)$$
where M_t_ and M_0_ are the gas concentrations in the polymer at saturation time t and the initial time, respectively, and $M_{\infty }$ is the solubility, which is the amount of gas in a full saturation state. L is the thickness of the specimen, D is the diffusion coefficient, and t is saturation time. Two experimental datapoints were required to calculate the diffusion coefficient and full saturation among the unknowns in Equation (2). When the saturation process was performed for 10 and 20 min, the weight gains were 7.87 and 11.09%, respectively. Calculated by substituting the above data into Equation (2), the solubility was 18.99%, and the diffusion coefficient was 8.8668 × 10^−3^ mm^2^/min under the saturation conditions of this study. In this study, to prevent gas adsorption of the polymer (PLA) jig, only the PMMA sample was put into the high-pressure vessel. After the saturation time, the specimen was taken out of the vessel and combined with the PLA jig under atmospheric pressure.

The tendency of weight gain during this process is shown in Figure 6. The x-axis represents saturation time, and the y-axis represents the weight gain calculated using Equation (1). To provide an appropriate difference in weight gain, the saturation times were selected as 10, 60, and 180 min, and the resulting weight gains were 6.13, 14.65, and 18.51%, which were confirmed to have a relative error within 0.2% compared to the value predicted by Equation (2).

### 3.2. Theoretical Mechanism

The conventional nucleation theory (CNT), which describes the formation of cells inside polymers, describes cell nucleation as a calculation of required critical work. The process of forming a polymer–gas single phase at high pressure increases the Gibbs free energy. To lower the Gibbs free energy, bubbles are voluntarily formed inside a polymer, and the change in the required work for the system is changed by increasing and decreasing the size of the bubbles, which can be expressed by Equation (3). Although various models are available for CNT, Equation (3) represents the change in work for homogenous models [33].
(3)$$\Delta W=-\left(P_{\mathrm{bub}}-P_{\mathrm{local}}\right)V_{\mathrm{bub}}+\gamma_{\mathrm{lg}}A_{\mathrm{bub}}$$
where W is the required work for bubble formation, P_bub_ is the bubble pressure, P_local_ is the local pressure, V_bub_ is the bubble volume, γ_lg_ is the surface tension between the bubble and the polymer–gas mixture, and A_bub_ is the surface area of the bubble. When the derivative of Equation (3) becomes zero, the corresponding change in work value can be regarded as the maximum value, that is, it can be calculated as the critical required work. The radius at that moment is determined as the critical radius and can be expressed by Equation (4) [34].
(4)$$R_{\mathrm{cr}}=\frac{2\gamma_{\mathrm{lg}}}{P_{\mathrm{bub}}-P_{\mathrm{local}}}=\frac{2\gamma_{\mathrm{lg}}}{P_{\mathrm{bub}}-\left(P_{\mathrm{sys}}+\Delta P_{\mathrm{local}}\right)}$$
where P_sys_ is the total system pressure, and ΔP_local_ is the local pressure fluctuation. P_local_ is considered as the internal pressure of the machine. Thus, P_local_ can be separated into P_sys_ and ΔP_local_ and can be expressed as the right term of Equation (4). If compression stress is applied to a specimen through this process, it has the same effect as tensile stress. Therefore, ΔP_local_ has a negative pressure, thereby reducing the critical radius, as shown in Equation (4). This implies that the probability of cell nucleation increases. Therefore, theoretically, compression pressure can play a role in increasing the probability of foaming. The weight gain and bubble pressure are linearly proportional according to Henry’s law. If the critical radius is sufficiently small due to the synergistic effect of compression stress and weight gain in the polymer, the activation energy for foaming is extremely small, resulting in foaming without thermal activation. When cells are created inside a polymer, a phenomenon occurs in which a transparent specimen appears opaque. This is because the path of visible light is interrupted by the cells inside the polymer and cannot penetrate. This is one of the drawbacks of foamed polymers, and research is currently underway to make foamed specimens transparent by forming sizes of cells that are smaller and more evenly spaced than 300 nm, the wavelength of visible light [35]. Images of the four specimens (neat, 10 min, 60 min, and 180 min) are shown in Figure 7, and we observed that the foaming process occurred at 180 min. A foaming process in which the cell was nucleated was conducted, so it was possible to visually check that the specimen turned opaque when saturated for 180 min compared to other specimens. It could be verified that a specimen that was not saturated (neat) was not patterned with compression molding under ambient pressure at room temperature. Additionally, it was found that the glass transition temperature was lowered through gas adsorption, so processing was possible, but the desired patterning could not be performed without nucleation of cells when the specimens saturated for 10 min and 60 min were checked. The quantitative analysis of the patterning is discussed in the following section through various microscope analyses.

### 3.3. Cell Morphology

We observed that a cell was formed inside the specimen with 18.51% weight gain after saturation for 180 min. To examine the internal structure, the cell morphology of the patterned material was confirmed using a field-emission-scanning electron microscope (JEOL, Model JSM-IT-500HR), and the images are shown in Figure 8. For the neat sample and samples with saturation processes performed for 10 and 60 min, no cells were generated. In contrast, a patterned part could be confirmed at 10 min and 60 min in which no cells were generated because gas molecules were dissolved into the polymer to reduce its chain bond, thereby lowering the glass transition temperature below room temperature [36,37]. The specimens with 10, 60, and 180 min of saturation time had maximum depths of 33.33, 129.5, and 208.7 μm, respectively. These values were 16.67, 64.75, and 109.7%, respectively, compared to the original patterning depth of 200 μm. The maximum depth of patterning was measured from the surface, and when cells were formed inside and foaming proceeded, patterning was successfully performed on a polymer of the desired depth at room temperature and ambient pressure. To confirm the exact cell morphology of the specimen subjected to saturation for 180 min, a scanning electron microscope (SEM) image with a higher resolution was captured. Figure 9 confirms that 1–10 μm cells were located between large cells of 200–300 μm, which can be identified in Figure 8. The cell density appeared to be lower than that of microcellular-foamed polymers, which generated cells through rapid phase changes using thermal activation energy. This was due to the lack of energy for cell growth. After the cell nucleation site was determined, it was determined through cell growth whether cells were formed or dissipated in the corresponding site. Additionally, if no external energy was present after the formation of cells, they coalesced into large cells to minimize surface energy.

The coalescence of the cells caused low cell density, which reduced the pressure generated between the cells and generated large cells with sizes from 200 to 300 μm for the aforementioned reasons. Due to the pressurization applied to the surroundings of the largely grown cells, geometry restrictions due to compression molding, and lack of energy for growth, small cells of approximately 1–10 μm were nucleated between the large cells at the same time. Foaming activation energy is energy required for the foaming process, and the process could proceed if the energy exceeded the foaming activation energy through energy transfer via compression stress and rapid temperature changes used in the microcellular-foaming process.

### 3.4. Surface Topology

The surface change through this process consisted of three stages: (i) macropatterning through the 3D-printing mold geometry, (ii) micropatterning by the layer height of 3D printing, and (iii) submicropatterning (roughness) by the volume expansion of cell nucleation. A schematic of this process is illustrated in Figure 10. A confocal laser scanning microscope (KEYENCE, Model. VK-X210) was used to observe the surface topology, and the results are shown in Figure 11. In the case of the neat specimen, it was a perfectly flat specimen. The top image in Figure 11 shows the surface, and the color map for the height of the specimen is shown at the bottom. In the case of the bottom image, the scale of the legend is different, indicating h_0_, h_1_, and h_2_. Each value is listed in Table 3. From the upper image, the patterning clearly performed better than other conditions at 180 min. The layer thickness of the 3D printer was 0.4 mm, and in the case of 180 min, the pattern was engraved every 0.4 mm in the layer-stacking direction. In the case of 10 min, only macroscale patterning by mold geometry was performed, and this depth was 32 μm, as measured by SEM image in Figure 8. In the case of 60 min, the pattern was engraved every 0.8 mm, and the dimension of the pattern was thicker and wider than intended. A graph plotted by measuring the height of the AA’ plane for each specimen is shown in Figure 12.

The tendency visually confirmed in Figure 8 can be verified numerically in Figure 12. Patterning rarely existed in a specimen after a saturation time of 10 min, and in the case of 60 min, the maximum depth was 130 μm, patterning was not performed properly, and unintended roughness occurred in a patterned part. This confirms that patterning did not proceed properly simply by pressing, and if a force was applied in the opposite direction to pressing, the patterning resolution was increased because volume expansion occurred through cell nucleation. The roughness was checked to verify the change in the surface, and the surface roughness of the entire region and the line roughness for the BB’ plane, which is perpendicular to the microscale pattern in Figure 11, were verified by selecting R_a_, R_q_, and R_p_ values, which are representative roughness values. The roughness data are listed in Table 4. R_a_ and R_q_ were calculated using Equations (5) and (6), respectively, where R_a_ is the arithmetic mean of the height, R_q_ is the root mean square of the height, and R_p_ is the maximum peak-sampling length. In the case of line roughness, we observed that the specimen with 60 min of saturation time had a roughness value from 4 to 8 times greater, and the 180 min specimen had a roughness value from 8 to 10 times greater compared to specimen with 10 min of saturation time.
(5)$$R_{a}=\frac{1}{l_{r}}\int_{0}^{l_{r}}\left|h\left(x\right)\right|\mathrm{dx}$$
(6)$$R_{q}=\sqrt{\frac{1}{l_{r}}\int_{0}^{l_{r}}h{\left(x\right)}^{2}\mathrm{dx}}$$

The increase in roughness compared to that of the specimen with a 10 min saturation time was due to the influence of microscale patterning, and the difference in roughness between 60 min and 180 min could be judged by the change in roughness based on the presence of the cellular structure in the polymer. The surface roughness for the entire region exhibited the same tendency, and 180 min exhibited a higher roughness parameter value because the macropatterning was deeper and better.

## 4. Discussion

This study proposed a process for patterning a solid-state polymer surface at room temperature and atmospheric pressure. A polymer–gas mixture was formed through the gas saturation process involved in the microcellular-foaming process, and patterning on three levels of scale on the surface of the polymer was achieved through compression molding and a jig patterned via a 3D printer. Cell morphology, surface-patterning resolution, and roughness were confirmed using a confocal laser scanning microscope and SEM, and as hypothesized, more accurate patterning could be performed if a force was applied in the opposite direction to the compression press with volume expansion by cell nucleation, as listed in Table 5. This implies that softening the polymer through gas adsorption did not allow accurate surface patterning, and the volume expansion force through cell formation inside the polymer increased the resolution of the patterning, confirming that the depth and clarity of a pattern can be controlled by controlling gas adsorption. 

In summary, a batch-foaming process was applied to a solid-state polymer to propose a process capable of geometric stamping and pattern imprinting while increasing roughness at the same time, and it was confirmed that it could be controlled through gas saturation time control. In addition, the reduction in energy consumption improved significantly compared to the conventional patterning process of compression molding. Compression molding is usually performed at high pressures (usually over 200,000 kPa) and at temperatures between 5 and 10 degrees above the glass transition temperature [38]. The process proposed in this study was carried out at much lower temperatures (room temperature) and lower pressures (less than 848 kPa) than normal compression molding, which was efficient and, at the same time, caused internal exchange (cell creation) through the microcellular-foaming process. This could lead to changes in various characteristics, such as surface characteristics and mechanical properties, and follow-up research on this study is essential. It is thought that various potential applications can be found through further research. Adhesion between polymers by foaming can be a potential application of this study. By using the process proposed in this study, it is expected that, when adhering between polymers, the binding force can be increased through increasing friction force (vertical drag) due to geometry matching and volume expansion, increasing the roughness between adhesive surfaces, as illustrated in Figure 13.

This study conducted only simple patterning. However, various studies are required for optimization, etc. through changes in structures used in actual bioscaffolds or microstructures, layer thickness of 3D printers, tool paths, etc. This process can be regarded as a novel application region for batch processes, which have received little attention from industry due to their low productivity. Given that various characteristic changes are possible through microcellular-foamed polymers, we expect that the solid-state surface-patterning process can be widely applied to various fields.

## Figures and Tables

**Figure 1 polymers-15-01153-f001:**
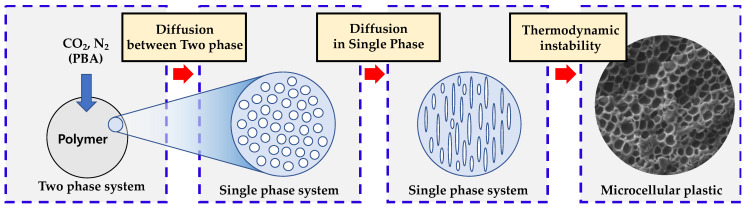
Mechanism of the microcellular-foaming process.

**Figure 2 polymers-15-01153-f002:**
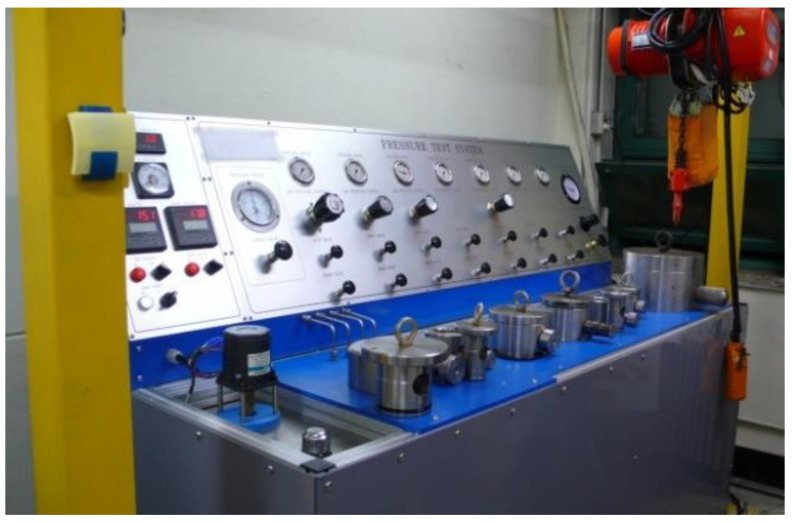
Experiment equipment for the solid-state batch-foaming process.

**Figure 3 polymers-15-01153-f003:**
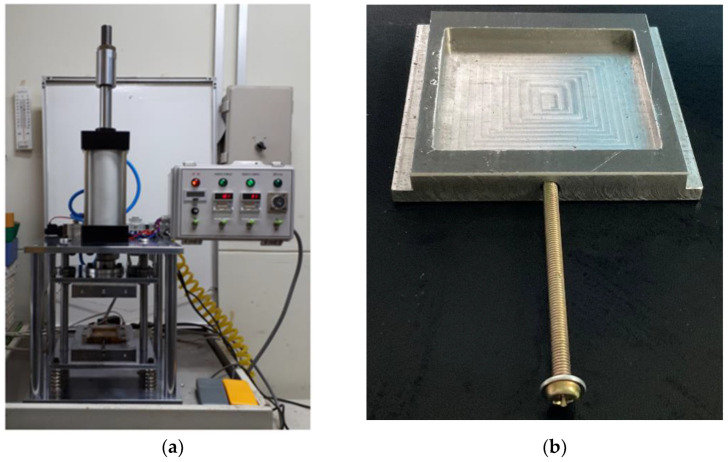
Experiment equipment: (**a**) compression molding and (**b**) compression-molding jig.

**Figure 4 polymers-15-01153-f004:**
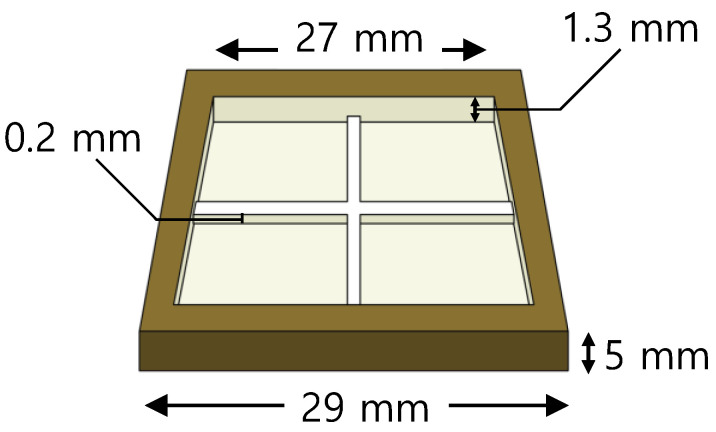
Schematic of jig for compression molding.

**Figure 5 polymers-15-01153-f005:**
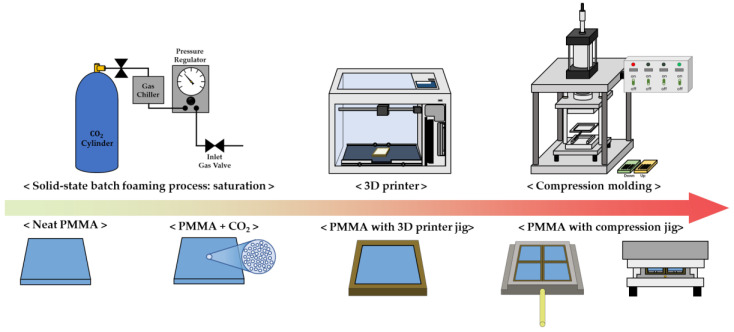
Schematic of overall process for this study.

**Figure 6 polymers-15-01153-f006:**
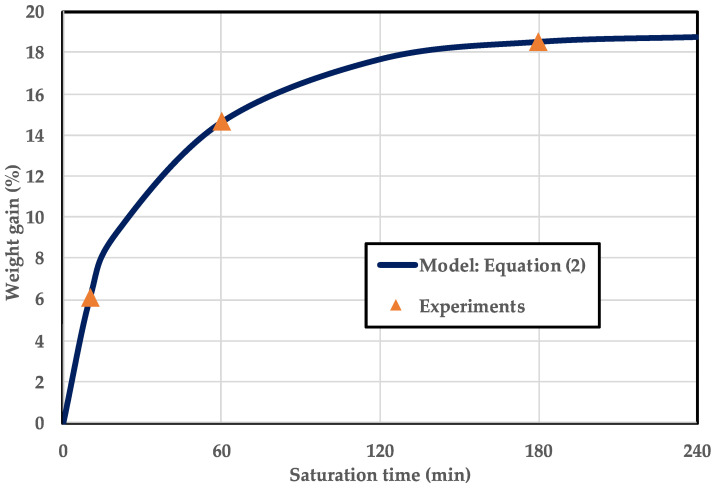
Gas adsorption curve as a function of saturation time at 5 MPa and 24 °C in PMMA.

**Figure 7 polymers-15-01153-f007:**
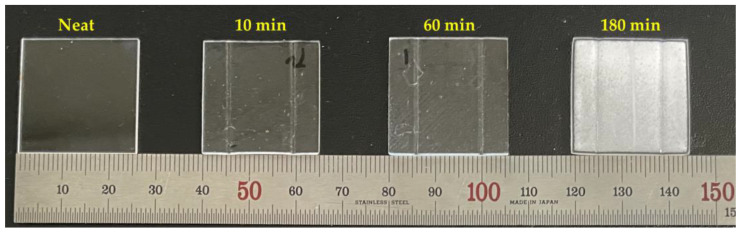
Image of the specimen after overall processing as a function of saturation time.

**Figure 8 polymers-15-01153-f008:**
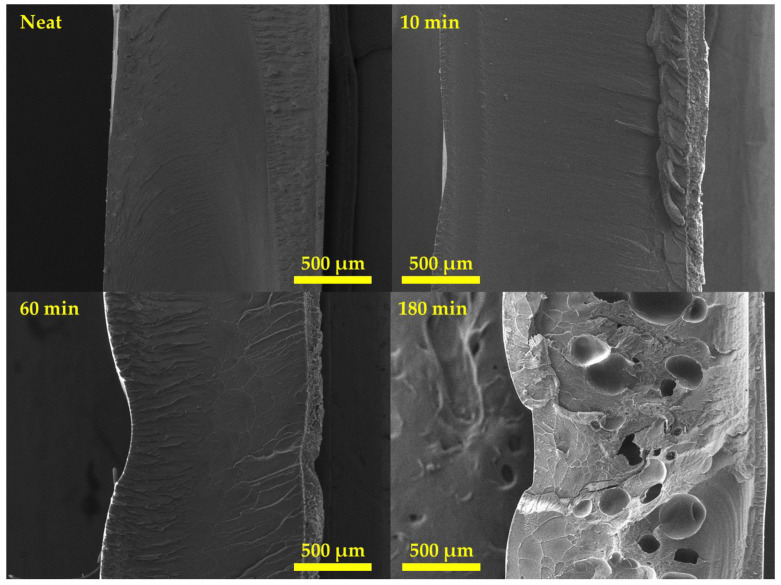
SEM images of specimen as a function of saturation time (×50).

**Figure 9 polymers-15-01153-f009:**
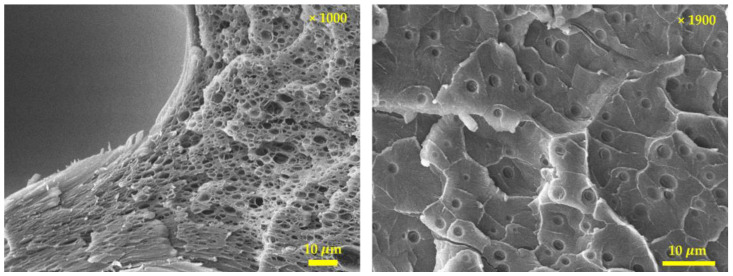
SEM images of specimen after solid-state surface patterning at 180 min of saturation.

**Figure 10 polymers-15-01153-f010:**
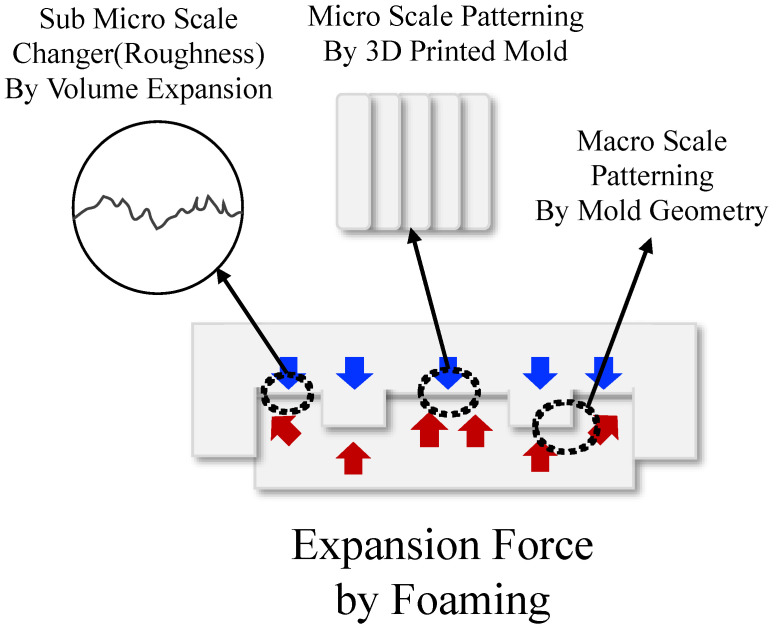
Schematic for the three-stage solid-state surface-patterning process.

**Figure 11 polymers-15-01153-f011:**
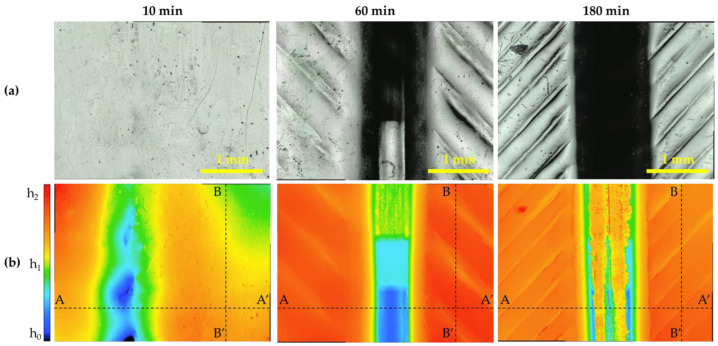
Images of confocal laser scanning microscope as a function of saturation time: (**a**) laser and optical; (**b**) color map for height.

**Figure 12 polymers-15-01153-f012:**
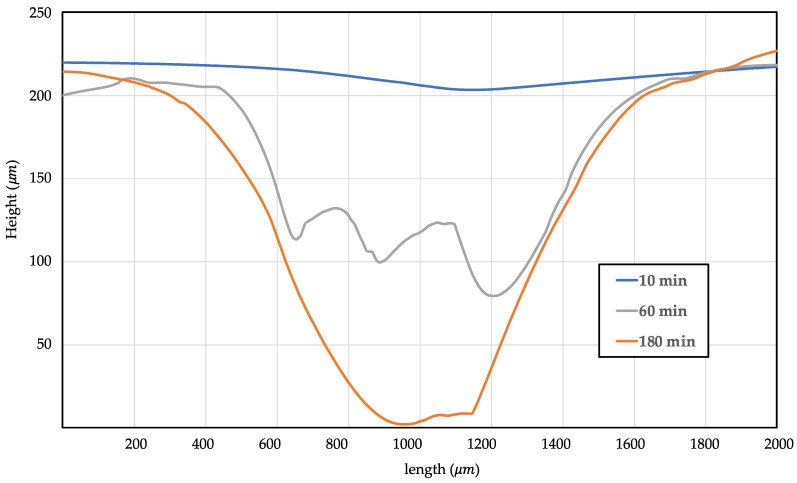
Height of macropatterning by compression stress of 3D-printing mold: AA’ line.

**Figure 13 polymers-15-01153-f013:**
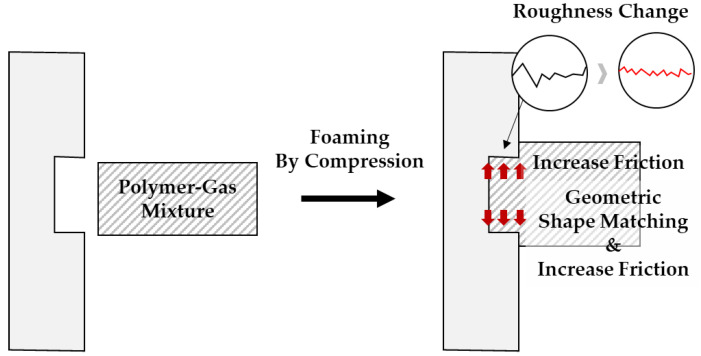
Adhesion between polymers by foaming as potential application.

**Table 1 polymers-15-01153-t001:** Specifications of the PMMA in this study.

Property	Value
Width × height (mm)	25 × 25 (±0.5)
Density (g/cm^3^)	1.19
Thickness (mm)	1.25–1.28
Purity (%)	99.98
Glass transition temperature (°C)	110
Melting temperature (°C)	158
Coefficient of thermal expansion (/°C)	6 × 10^−5^

**Table 2 polymers-15-01153-t002:** Experimental parameters used in this study.

Property	Value
Saturation pressure (MPa)	5.5 (±0.05)
Saturation temperature (°C)	20 (±0.5)
Saturation time (min)	10/60/180
Depressurization ratio (MPa/s)	5.5
Foaming temperature (°C)	-
Foaming time (min)	-

**Table 3 polymers-15-01153-t003:** Height parameters of legend in Figure 11.

Parameters	10 min	60 min	180 min
H_0_	25	0	0
H_1_	45	125	150
H_2_	65	250	300

**Table 4 polymers-15-01153-t004:** Surface roughness parameters of all regions and BB’ line in Figure 11.

Parameter	10 min		60 min		180 min	
	All Regions (μm)	Line (μm)	All Regions (μm)	Line (μm)	All Regions (μm)	Line (μm)
R_a_	4	2	29	9	42	22
R_q_	6	5	40	23	57	37
R_p_	14	3	65	24	98	39

**Table 5 polymers-15-01153-t005:** Summary of experiment results of research.

Saturation Time	Weight Gain	Maximum Depth		Pattern Gap		Roughness (All Regions)		
		Mold	Sample	Mold	Sample	R_a_ (μm)	R_q_ (μm)	R_p_ (μm)
10 min	6.13%	200 μm	335 μm	0.4 mm	-	4	6	14
60 min	14.65%		129 μm		0.8 mm	29	40	65
180 min	18.51%		208 μm		0.4 mm	42	57	98

## Data Availability

Not appliable.

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
