# Peer review of "Solid-State Surface Patterning on Polymer Using the Microcellular Foaming Process"

_polymers, 2023, doi:10.3390/polym15051153_

Round 1

Reviewer 1 Report

The topic discussed in the work is up-to-date and very important.

The work is very concise, there are  imperfections and simplifications that should be corrected.

And thus analyzing:

The abstract should include the specific results discussed in the article

Part of the research methodology should be described in more detail.

Research apparatus, machines and devices are not described.

Which of the drawings and diagrams are the authors', and which are from the literature.

If the figures are not the author's, permission must be obtained and relevant literature cited.

The results should be also summarized in tables, which will allow for their interpretation.

The analysis of the test results should indicate and explain the test results.

No summary was made and conclusions were not precisely defined.

Reviewer 3 Report

The manuscript presents a solid-state process for creating micro-patterns on polymers. Although the research has been presented with sufficient details, the background of the work is not properly presented. The authors should take care of the following aspects to improve the manuscript-

1. Micro-pattering by compression moulding is a fairly common technique, and the authors should establish the novelty of the work.

2. Figure 7 needs clarification.

3. 3D printed surfaces have undulations because of the process characteristics, and such undulations are not very uniform. Did the authors consider those undulations as the micro patterns on the moulds?

4. Did the authors use any mould-release agent.

5. Can the authors justify the works with some potential applications?

Reviewer 4 Report

1. In the "Abstract", line 10, what is "MCPs" short for, since this is its first appearance.

2. In the "Introduction", from line 71 to line 79, I am still confused what is the problem this manuscript trying to solve? What are its advantages over previous studies?

3. Figure 2 has never been referenced in the text, please double check.

4. The length sacle in Figure 8 and Figure 9 are “m”, these could not be right.

5. How to distinguish the subfigures in Figure 11, please mark them with right numbers.

6. Please doule check the contents from Line 341 to Line 344, this should not be a draft.

7. I would like to suggest going through the manuscript more carefully for clarity, syntax and correctness. The English should be improved for the sake of clarity.

Round 2

Reviewer 2 Report

It is fine after this careful revision.

Reviewer 3 Report

The authors have addressed the queries adequately, and the manuscript is acceptable in its current form.

Reviewer 4 Report

Accept